# Changes in Breathing Patterns after Surgery in Severe Laryngomalacia

**DOI:** 10.3390/children8121120

**Published:** 2021-12-03

**Authors:** Fabrizio Cialente, Duino Meucci, Maria Luisa Tropiano, Antonio Salvati, Miriam Torsello, Ferdinando Savignoni, Francesca Landolfo, Andrea Dotta, Marilena Trozzi

**Affiliations:** 1Airway Surgery Unit, Pediatric Surgery Department, Bambino Gesù Children’s Hospital, 00165 Rome, Italy; duino.meucci@opbg.net (D.M.); marialuisa.tropiano@opbg.net (M.L.T.); antonio.salvati@opbg.net (A.S.); miriam.torsello@opbg.net (M.T.); marilena.trozzi@opbg.net (M.T.); 2Neonatal Intensive Care Unit, Department of Neonatology, Bambino Gesù Children’s Hospital, 00165 Rome, Italy; ferdinando.savignoni@opbg.net (F.S.); francesca.landolfo@opbg.net (F.L.); andrea.dotta@opbg.net (A.D.)

**Keywords:** laryngomalacia, stridor, epi-glottoplasty, pulmonary function test, airway obstruction, endoscopy, lung function test

## Abstract

Background: Most of the studies regarding the surgical treatment of severe laryngomalacia (LM) have been aimed at describing the efficacy of the treatment in terms of improvement of clinical symptoms or anatomical findings. There are no studies specifically aimed at analyzing the changes in breathing patterns following surgical treatment for severe LM. Objective: To review the breathing pattern changes before and after corrective surgery in infants with severe LM. Study design: A series of retrospective cases at a tertiary referral children’s hospital. Methods: Retrospective chart review of 81 infants who underwent supra-glottoplasty (SGP) for severe laryngomalacia between 2011 and 2020 at Bambino Gesù Children’s Hospital of Rome, Italy. Among the patients, 47 (58%) were male and 34 (42%) were female. Twenty-one patients (26%) had one or more comorbidities condition. The data collected included age, symptoms, a polysomnography/pulse oximetry study, growth rate, the findings from flexible endoscopy, pre- and post-supra-glottoplasty (SGP) pulmonary function tests (PFTs) and, when indicated, 24 h pH-metry. Breathing patterns were studied during restful, normal sleep, using an ultrasonic flow-meter (Exhalyzer, Viasys) which measured: Tidal Volume (Vt), Respiratory Rate (RR), time to peak expiratory flow/expiratory time ratio (tPTEF/Te, an index of the patency of the lower airways) and mean expiratory/mean inspiratory flow ratio (MEF/MIF, an index of the patency of the upper airways) evaluated before surgical procedure (T1) and 3–6 weeks after (T2). Pre- and post-operative mean data were calculated and comparisons made with a Student *T*-test. Results: The surgical procedure was well tolerated by all infants and no intraoperative or post-operatory long-term complications were noted. In T1, breathing patterns were characterized by low tidal volume and high tPTEF/Te and MEF/MIF ratios, suggesting a severe reduction in the patency of the upper airways in all patients. After surgery (T2), all the previously mentioned variables significantly improved, reaching normal values for the child’s age. Conclusions: Supra-glottoplasty, as already described in several studies, is a safe and efficient procedure to treat severe laryngomalacia during infancy. The improvement in breathing patterns after surgery was reliably confirmed by a lung function test in our study, which showed the diagnostic value of testing respiratory functionality in the laryngomalacia and comparing them to clinical and endoscopic data. Moreover, considering the results obtained, we also propose the use of this available, dependable test to verify its therapeutic effects (post-surgery) and to monitor future respiratory development in these infants. Moreover, we believe that further studies will provide detailed grading guidelines for gravity of the LM, based on these functional lung tests.

## 1. Introduction

Laryngomalacia (LM) is the most common (60%) congenital laryngeal anomaly and the first cause of stridor in children under 2 years of age, accounting for approximately 65–75% of all cases of stridor [1,2]. Of unknown etiology, multiple causal theories have been proposed, with neurologic dysfunction as one of the leading. In any case, LM is characterized by inspiratory prolapse of the supraglottic structures into the airway due to a weak laryngeal tone (Figure 1).

While previous studies have suggested that males were affected twice as often as females, recent evidence suggests that it is equally common in both genders. No racial differences have been reported, although black and Hispanic infants seem to be at an increased risk compared to white infants [3].

Although the exact pathophysiology is still unknown, different causal theories of LM have been proposed. These include neurological and anatomical explanations, but neurologic dysfunction is one of the leading theories: this last theorizes that neuro-sensorimotor dysfunction leads to decreased coordination and neuromuscular tone of the laryngeal structures, resulting in dynamic collapse of the supraglottic structures into the airway during inspiration [4]. This concept is supported by study showing submucosal nerve hypertrophy in a histological specimen of supraglottic tissue in an infant with severe LM [5]. Though the relationship between laryngomalacia and gastroesophageal reflux has been well documented, the causal relationship remains unclear [4,5,6]. One possibility is that gastric reflux stimulates an inflammatory response in the laryngeal mucosa, causing tissue edema which leads to airway obstruction. Another school of thought holds that children with laryngomalacia generate unusually large negative intrathoracic pressure gradients during inspiration when trying to overcome supraglottic airway obstruction, with consequent gastric reflux [5,6].

LM is characterized by high pitched inspiratory stridor beginning within 2 weeks of birth. The stridor usually worsens with agitation, crying, supine position and feeding, its intensity frequently fluctuating and variable according to the changes in the child’s body position [6].

A generally self-limiting condition, resolving itself by two years of age, LM may occur as isolated, or in association with other anomalies in the airway tract as in other organic systems, such as gastroesophageal reflux disease, craniofacial anomalies, and neurocognitive condition: children with LM, especially those with moderate and severe disease, have a 7.5–64% chance of having a synchronous or secondary airway lesion [5,7].

In most cases, infants between 18 and 24 months display a spontaneous improvement of the symptoms described [8], nevertheless in 5–10% of subjects the symptoms are more severe resulting in feeding difficulties, failure to thrive, apnea, or pectus excavatum [9,10]. Furthermore, so called late-onset laryngomalacia may also occur in infants older than 2 years without a history of prior disease [6,7].

Diagnosis of LM is established by an awake flexible nasopharyngoscopy that allows direct visualization of the upper aerodigestive tract (oropharynx, supra-glottis, glottis, sub-glottis, and hypopharynx) during respiration. The endoscopic appearance of LM is often a combination of long, curled (omega-shaped) retro-positioned epiglottis, shortness of aryepiglottic folds, and bulky arytenoid prolapsed forward on inspiration. In the most severe cases, the epiglottis prolapses posteriorly to the glottic level on inspiration, causing a complete closure of the airways [11]. Direct laryngoscopy and diagnostic bronchoscopy in the operating room, under general anesthesia and spontaneous ventilation, give a full evaluation of the trachea to the level of the mainstream bronchi. This procedure is the mainstay in patients with severe symptoms or in case there is suspected synchronous airway lesions [12]. Differential diagnosis includes unilateral or bilateral vocal fold paralysis, subglottic stenosis/hemangioma, trachea- or bronco-malacia, vascular ring, laryngeal papillomatosis and foreign body aspiration [12].

Although treatment of the majority of patients with LM is conservative, infants with moderate or severe stridor, recurrent cyanosis, apneic pauses with moderate-severe OSA (Apnea Hypopnea Index AHI > 5–10), feeding difficulties with weight loss, failure to thrive, or severe suprasternal and subcostal retraction warrant immediate surgical intervention [13,14,15].

In the past, tracheostomy was the standard of care [15,16,17]. Recently, endoscopic supra-glottoplasty has become the treatment of choice for patients with severe LM [16,17,18,19]. Various studies describe different surgical techniques for this procedure, including the use of cold steel instruments, CO_2_ laser, laryngeal micro-debrider or coblator [14,15,16,17,20,21]. Regardless of the technique used, supra-glottoplasty consists of dividing the short aryepiglottic folds, resulting in epiglottic release, and removal of redundant and prolapsing tissue in the supra-epiglottic region [19,20,21].

In our experience pulmonary function testing (PFT) plays an important role in the evaluation of the infant with known or suspected LM, allowing for assessment of the type and degree of obstruction and the impact of therapies. As already described in several studies, the shape of the flow–volume curve shows typical characteristics in specific ventilation disorders. In fact, the expiratory part of the curve is concave in obstructive respiratory diseases, such as laryngomalacia, while in restrictive disorders the shape of the curve is without pathological changes but with overall reduced values. PFT can also distinguish fixed from variable obstruction during the different respiratory phases.

Most studies examining the effectiveness of surgical treatment of LM refer to primary outcome measures, such as resolution of stridor and/or improved growth, and in terms of the improvement of anatomical findings [19,20,21,22]. There are no studies specifically designed to analyze the changes in breathing patterns following any surgical treatment for severe LM.

According to our experience, infants with noisy breathing can be evaluated for their respiratory function by using tidal-breathing flow-volume loop analysis (TBFV). This non-invasive and effective method can help in diagnosing the site and degree of airway obstruction [21,23,24,25].

Given the shortage of clinical data in studies which support the use of such respiratory tests, we have conducted a pre- and post-operative comparative study in a series of infants with severe LM who underwent AEP surgery.

Starting from our experience of feasibility of using LFT in children and infants with suspected congenital thoracic arterial anomalies (CTAAs) [26], the aim of this work was to highlight the reliability of these respiratory tests in a series of diagnostic procedures, in the evaluation of the therapeutic results after supra-glottoplasty, and in the follow-up and future monitoring of respiratory patterns.

## 2. Methods

From January 2011 to March 2020, 320 infants with chronic stridor were evaluated at our tertiary referral children’s hospital Bambino Gesù, Rome, Italy, on the basis of a multi-disciplinary protocol including: clinical evaluation, polysomnography, pulmonary function tests (PFT), airway endoscopy and, when required, 24-h pH-metry.

For every patient, the diagnosis of severe LM had been made based on pre-operatory airway endoscopy, performed via two methods: first, an awake naso-fibroscopy with flexible naso-pharyngoscope (Olympus 3.5 mm) showed a complete view of the nose, rhino-oro-hypopharynx, the supra-glottis and the glottis, to obtain a clear assessment of airway dynamics. Next, a direct laryngoscopy and diagnostic bronchoscopy with a 4-mm 0°-degree rigid telescope is used for a complete and accurate exploration of the laryngeal and tracheobronchial tract under general anesthesia and spontaneous ventilation, without using myorelaxant drugs. The direct viewing of the sovra-glottic collapse during inspiration leads to the diagnosis of LM.

Lung function was assessed by analysis of the tidal-volume and flow-volume loop, using an ultrasonic flow-meter (Exhalyzer, Sensor Medics). PFTs were performed at our children’s hospital in neutral supine position during normal sleep, without any sedation, recording at least 20 consecutive breaths of each patient, while also measuring the following parameters: tidal volume (Vt, mL/kg), respiratory rate (RR, a/m’), and mean expiratory/mean inspiratory flow ratio (MEF/MIF). PFTs were performed before surgery (T1) and 3–6 weeks after (T2).

Eighty-one cases were diagnosed as severe laryngomalacia, requiring surgery. Severe LM was defined as the presence of specific endoscopic findings (severe inspiratory prolapse of the supraglottic structures into the airway) associated with airway symptoms (apnea, cyanosis or pectus excavatum), intractable feeding difficulties/weight loss, and failure to thrive. The exclusion criteria for surgery were the presence of neurological conditions, such as cerebral palsy, hypotonia, mental retardation and developmental delay, because of the evidence of significantly higher rate of failure of supra-glottoplasty in patients with neurological disease, as reported in the literature [27,28].

Patients included in our work were studied at a post-natal age between 1 and 8 months (median age at surgery = 3.5 months).

All patients underwent endoscopic bilateral supra-glottoplasty performed using cold laryngeal micro-instrumentation. A direct laryngo-tracheobronchial endoscopy was performed under general anesthesia using a vallecular laryngoscope gently introduced for a clear view of the larynx, particularly the aryepiglottic folds and the arytenoids. The patient was in the supine position with the head slightly extended to optimize the exposition. A nasopharyngeal cannula was placed for oxygenation. Using a sinuscope 0°-degree 4 mm, in suspension or held by an assistant surgeon, cold steel instruments (micro-laryngeal scissors) were used to incise the aryepiglottic folds permitting the immediate distension of the epiglottis. Then, the redundant sovra-arytenoid mucosa was cut and removed with micro-laryngeal graspers and scissors. The same treatment was repeated on the opposite side. During the surgical procedure intermittent endotracheal intubation may be required if desaturation occurs. Care must be taken to prevent inter-arytenoid scarring and subsequent supraglottic stenosis.

Following the operation, the patient retained the tubing and was placed on a respirator for 24 h in the PICU (Pediatric Intensive Care Unit). We kept the tubing during the 18–24 h after surgery to protect the airway in case of post-operative bleeding and to help the patient breathe properly in the case of an extreme edematous reaction. Furthermore, we believe this could help to reduce the risk of supraglottic scar. Three-four weeks later PFTs were assessed and a laryngeal endoscopy was performed (Figure 2 and Figure 3).

The endoscopy was video recorded in each phase to allow a multi-disciplinary diagnosis and a video comparison was made of the results after surgery.

Statistical analysis was performed using the student’s *T*-test. The mean differences between the T1 and the T2 tests were considered statistically significant for “*p*” values < 0.05.

## 3. Results

Eighty-one patients met the criteria for inclusion in this study. The gender distribution included 47 males (58%) and 34 females (42%). The mean-age at the time of surgery was 3.5 months (with a range of 1–8 months). The most common presenting comorbidity was gastroesophageal reflux (90%), diagnosed based on the presence of clinical symptoms and confirmed by esophageal-pH-monitoring. Indications for surgical procedure in our cohort included: persistent respiratory distress, feeding difficulties with weight loss, severe suprasternal and subcostal retraction, and failure to thrive.All 81 patients underwent an endoscopic supra-glottoplasty with the tube successfully removed 18–24 h after surgery. The operation was well-tolerated by all patients and no complications or negative outcomes, such as supraglottic stenosis or dysphagia, were detected in the immediate and tardive post-operative situation. All cases showed a good surgical outcome with a complete resolution of stridor breathing and disappearance of feeding difficulties after SGP.

In the initial days after the tube was removed, the patients began to take nourishment immediately without complications. The total length of follow-up post-surgery ranged from 1 weeks to 2 years.

In all cases, airway fibro-laryngoscopy performed at 4 weeks after surgery displayed a good cicatricial process at the supraglottic level, with no signs of retraction (Figure 3). All pulse oximetry/polysomnographic parameters had become normal and desaturations disappeared. When performed, comparison of pre- and post-operative polysomnography revealed a decrease of 75% in the mean value of the obstructive apnea index (OAI) and an increase of 13% in the mean low- arterial-oxygen-saturation (SpO_2_ nadir).

The average time between the pre-operative pulmonary evaluation and surgery was 2 weeks (range 8–35 days). Four-six weeks after surgery (range 2–12 weeks) each patient again received a lung function evaluation. The first (T1) PFTs were performed at a mean post-natal age of 2.9 ± 1.7 months; the second (T2) test was performed at a mean post-natal age of 4.5 ± 1.9 months. Comprehensive details of the pre- and post-operative evaluation of breathing patterns are included in Table 1.

The results showed a significant improvement after surgery in tidal volume (Vt pre = 5.5 ± 1.8 mL/kg; Vt post = 8.0 mL/kg) and the MEF/MIF ratio (MEF/MIF pre = 1.59 ± 0.77; MEF/MIF post = 0.85 ± 0.23), all of which were statistically noteworthy (*p* < 0.001). No significant differences in RR were observed between T1 and T2.

Overall, we observed a noted improvement in ventilatory patterns due to an increase in tidal-volume and inspiratory-flow, as a result of the drop in airway resistance during the breathing process (Figure 4).

## 4. Discussion

First described by Jackson in 1942, laryngomalacia is the main cause of congenital stridor in children accounting for almost 60–70% of cases, but its etiology is still a matter of debate. Often considered a self-limiting disease that resolves by 12–24 months of age, approximately 5–20% of patients need some sort of surgical intervention. The diagnosis of LM is established by direct visualization of the larynx via endoscopy.

Many studies have shown that supra-glottoplasty is an effective treatment in case of severe LM, and its benefits in a variety of outcomes measures with low complication rates have been reported in the literature. Main indications for endoscopic supra-glottoplasty are growth deficit due to feeding difficulties, plasma oxygen desaturation and/or moderate-severe OSA (AHI > 5–10). According to various sources of evidence, obstructive sleep apnea (OSA) may contribute significantly to laryngomalacia severity and may help guide decisions for surgical intervention [29].

Most of the previous studies regarding the surgical treatment of LM have been aimed at describing the effectiveness of the treatment in terms of improvement of clinical symptoms or anatomical findings. There are no studies specifically designed to analyze the changes in breathing patterns following surgical treatment for severe LM.

At present, the criteria for evaluating LM are exclusively clinical, endoscopic and based on few objective tests and variables and on a physician’s sensibility and experience [29,30,31]. In evaluating the effectiveness of the choice of treatment in cases of severe LM, some authors have proposed the use of polysomnography (PSG) [9]. The test indicates obstruction of the airway causing apneic or hypopneic pauses but, according to a recent study, this alone is not able to determine the level of obstruction, the severity of laryngomalacia or prediction of which patients would require surgical intervention [31,32,33,34].

In our study we focused on the changes in breathing patterns following supra-glottoplasty in infants with severe isolated LM, defined as the presence of feeding difficulties, breathing symptoms and failure to thrive. Thus, our decision to perform surgery was not based on endoscopic classification of LM, but on its clinical severity.

Improvement in breathing patterns after surgery was reliably confirmed by a lung function test. In particular the results showed a significant improvement (*p* < 0.001) after surgery in tidal volume and the mean expiratory/mean inspiratory flow ratio (index of the patency of the upper airways). Supra-glottoplasty, as already described in other studies, is a safe and efficient procedure to treat severe laryngomalacia during infancy [25,26,30,31,35]. This procedure is an effective technique which results in a significant improvement in breathing patterns.

Considering our results, we believe that a lung functional test could be used, in association with airway endoscopy, as a definitive diagnostic instrument for preoperative clinical staging in case of severe LM, as well as in the follow-up management in cases in which a surgery option is not required.

## 5. Conclusions

Laryngomalacia (LM) is the most common (75%) cause of stridor in infants. Considered a self-limiting condition, in most cases the disease follows a benign course. However, in 10–15% of cases the prognosis is less favourable, often requiring surgical intervention. Airway endoscopy, performed with flexible and rigid instruments, is the only definitive way to confirm or rule out the presence of LM in children presenting with a complaint related to the airway system.

With this study, we have underlined the diagnostic value of the pulmonary function tests in LM, comparing them with clinical and endoscopic data. We also recommend the use of these readily available, reliable tests to verify the therapeutic effectiveness (post-operative) and to follow-up these infants in their respiratory development. Finally, we underline the importance of a multi-disciplinary approach for infants with persistent stridor to identify the most severe cases and prescribe proper corrective surgery.

## Figures and Tables

**Figure 1 children-08-01120-f001:**
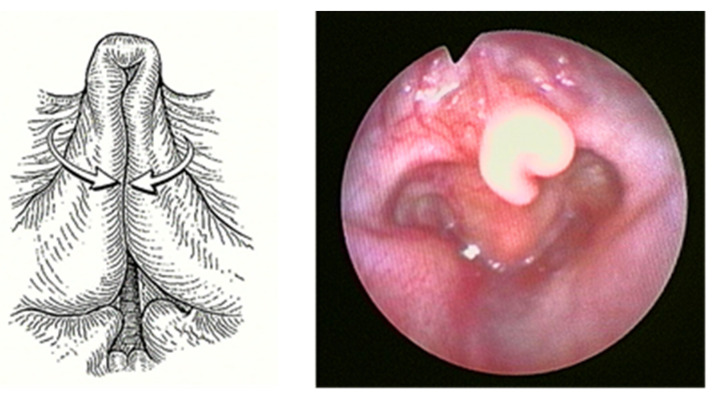
Severe Laryngomalacia.

**Figure 2 children-08-01120-f002:**
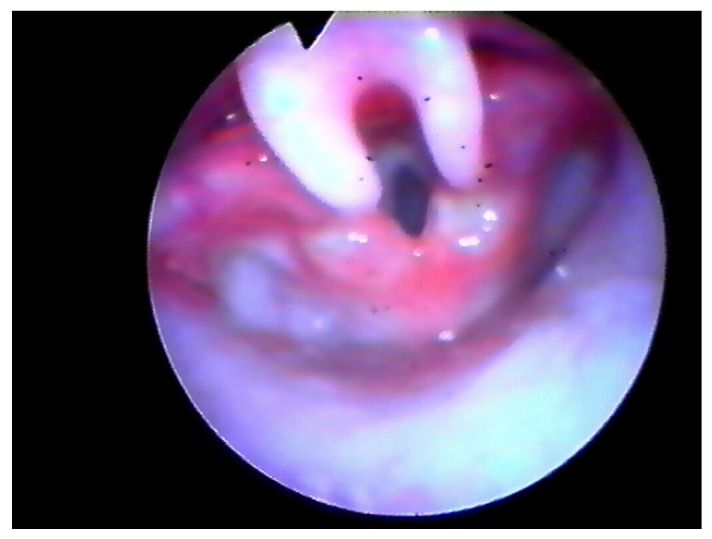
Endoscopic check, 8 days after surgery.

**Figure 3 children-08-01120-f003:**
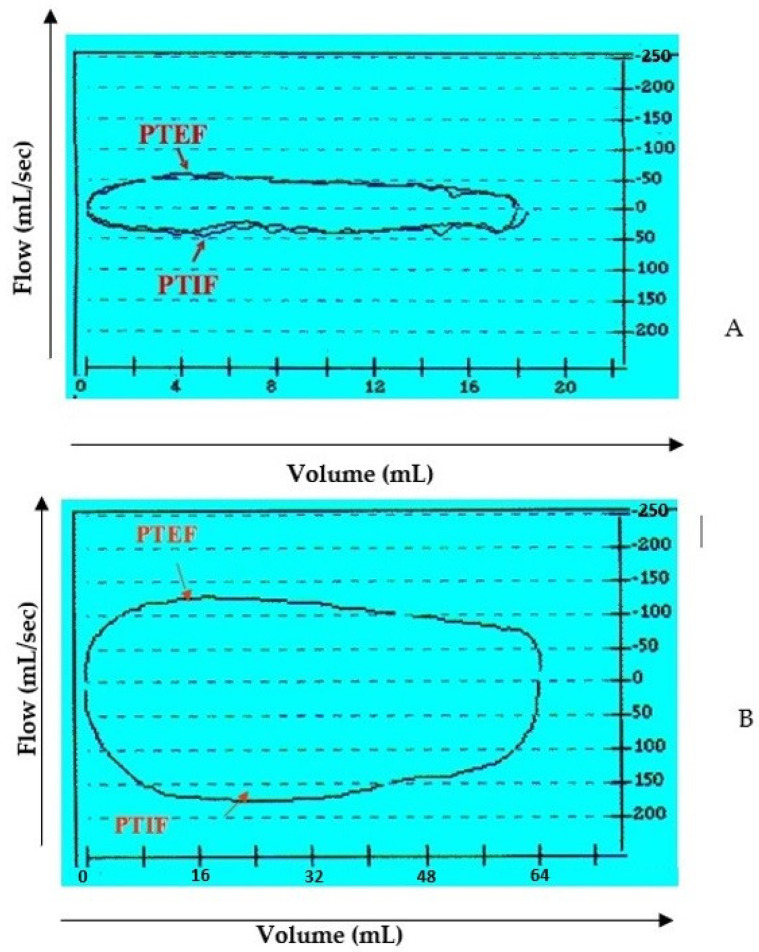
Flow-volume loop at tidal breathing. Example between a preoperative (**A**) and post-operative (**B**) case. Definition of abbreviation: PTEF = peak tidal expiratory flow; PTIF = peak tidal inspiratory flow.

**Figure 4 children-08-01120-f004:**
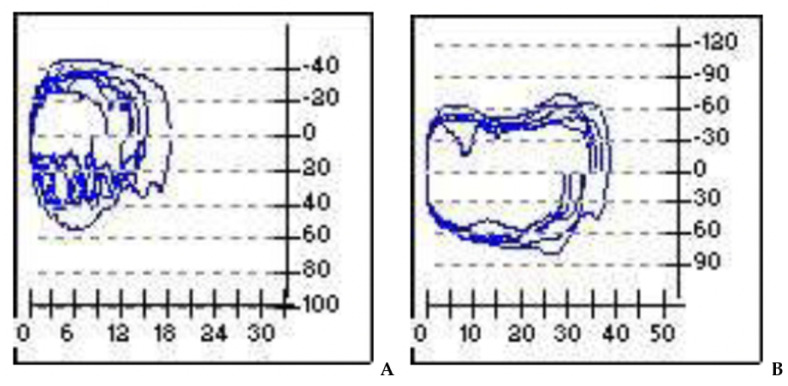
Pre- and post-operatory. (**A**): in the flow volume and flow time curves evident inspiratory obstruction; (**B**): overall improvement of the obstruction after surgery. X-axis = volume (m/L); y-axis = flow (mL/s).

**Table 1 children-08-01120-t001:** Changes in breathing patterns after supra-glottoplasty in infants with severe laryngomalacia.

	T1	T2	*p*
Tidal Volume, Vt (mL/kg)	5.5 ± 1.8	8.0	<0.001
Mean Expiratory Flow/Mean Inspiratory Flow (MEF/MIF) Ratio	1.59 ± 0.77	0.85 ± 0.23	<0.001
Respiratory Rate (Breathes/min)	53.7 ± 12.33	48.2 ± 15.3	NS

## Data Availability

Data were retrospectively analyzed in line with personal data protection policies.

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
