# Peer review of "Changes in Breathing Patterns after Surgery in Severe Laryngomalacia"

_children, 2021, doi:10.3390/children8121120_

Round 1

Reviewer 1 Report

This is a very nice paper.  There are several minor grammatical and spelling errors (eg page 3 line 1, page 7 3rd line of discussion). In the bigger picture, that is really not important.  

This paper makes an excellent effort at showing the benefit of supraglottoplasty on improving breathing patterns in children with severe laryngomalacia.  However some things were not clear to me.  Was this asleep or awake (and did that matter).  If asleep, and if this is more analogous to a sleep study, then it would be worth referencing the laryngomalacia sleep literature  in infants (Dave White was the first to lead in this area I think).  

Additional comments:  Table 1 is the the single most critical aspect of this paper, and it has a critical typo - T1 should be 5.5, not 55!

Also Figure 2 should be placed between figures 1 and 3

Author Response

Dear Colleague. Thank you for your suggestions. I rewied our paper according to your request. Waiting for your response, best regards and thank you again.

Fabrizio Cialente

Reviewer 2 Report

1. Please address or state comorbidities except gastroesophageal reflux. Do comorbidities include neurologic disorders or other synchronous airway lesions? These comorbidities usually have negative impact on LM and the surgical outcome of SGP, did you also have less favorable outcome in these subjects? 2. In this article, the author mentioned comparing PFT with endoscopic data, but usually in endoscopic finding the LM is divided into three types, did you record or compare LM types and PFT results? 3. For PFT, could you recognize the obstruction site, for example, supraglottis, glottis, and subglottis? If not, what’s the superior advantages of PFT than rigid bronchoscopy and flexible bronchoscopy for diagnosing LM? Or could you predict the severity of LM from PFT? 4. How did you define “severe” LM? By endoscopic finding for morphology or clinical finding? 5. Please clarify your surgical procedure. What does AEP stand for? 6. Do figure 2 and 4 belong to same patient? Could you provide comparison of PFT figure of same patient before/after surgery?

Author Response

Dear collegue. Thank you for your advices. Hope the answers will satisfy your request.

Waiting for news, best regards. Thank you again

Fabrizio Cialente

Round 2

Reviewer 2 Report

1.For your reply, we still have to address that PFT can’t distinguish the obstruction site of supraglottis, glottis, and subglottis. Thus, flexible bronchoscopy and rigid bronchoscopy are still warranted for diagnosing LM and survey for secondary airway lesion. For your article, you can emphasis the advantage of using PFT to evaluate surgical outcome, but it may not provide better value of diagnosis of LM.

2.For conclusion part, since you don’t distinguish the endoscopic types of LM, you may not compare the severity of LM from endoscopy with PFT results.  

Author Response

Dear colleague,

First of all, I would like to thank you for your review. About the first point, we added in the conclusion that “we believe that lung functional test could be use, in association with airway endoscopy, as definitive diagnostic instrument for preoperative clinical staging in case of severe LM, as well as in the follow-up management in case of surgery option is not required”. The stadiation is established by direct visualization of the larynx on endoscopy but the advantage of using PFT is that the test is a fast, unexpensive  and non-invasive and it can be used to define the mechanism of respiratory failure, improving the treatment and its effect, and is therefore a useful tool in the follow-up management for both surgical and non-surgical cases of newborn and infant with polmonary disease. It also spares the patient from a general anesthesia.